# Mice Overexpressing Wild-Type *RRAS2* Are a Novel Model for Preclinical Testing of Anti-Chronic Lymphocytic Leukemia Therapies

**DOI:** 10.3390/cancers15245817

**Published:** 2023-12-12

**Authors:** Alejandro M. Hortal, Ana Villanueva, Irene Arellano, Cristina Prieto, Pilar Mendoza, Xosé R. Bustelo, Balbino Alarcón

**Affiliations:** 1Immune System Development and Function Program, Centro Biología Molecular Severo Ochoa, Consejo Superior de Investigaciones Científicas (CSIC), Universidad Autónoma de Madrid, 28049 Madrid, Spain; avillanueva@cbm.csic.es (A.V.); iarellano@cbm.csic.es (I.A.); cprieto@cbm.csic.es (C.P.); pmendozadaroca@gmail.com (P.M.); 2Centro de Investigación del Cáncer, Instituto de Biología Molecular y Celular del Cáncer and Centro de Investigación Biomédica en Red de Cáncer, Consejo Superior de Investigaciones Científicas (CSIC), Universidad de Salamanca, 37007 Salamanca, Spain; xbustelo@usal.es

**Keywords:** R-RAS2, RAS, GTPases, chronic lymphocytic leukemia, B-CLL, mouse model, ibrutinib, venetoclax, new therapies, cancer treatment

## Abstract

**Simple Summary:**

Chronic lymphocytic leukemia (CLL) is a prevalent blood cancer, more common in men in their sixties or older. Unlike some other cancers linked to the KRAS gene, CLL is associated with a gene called RRAS2, which is overly active but not mutated. In a significant discovery, we established a mouse model by increasing RRAS2 levels, resulting in CLL development in 100% of cases. This model is essential for testing potential treatments before human trials. In this study, we validated the mouse model by evaluating two widely used CLL drugs, ibrutinib and venetoclax, and found that they effectively killed leukemia cells in the mice. This validation indicates that the mouse model can reliably simulate CLL responses to established drugs. Importantly, it opens the door to testing novel drugs, including those targeting RRAS2, which has not been explored in clinical settings. This advancement is a crucial step toward identifying improved therapies for CLL and holds promise for future developments in leukemia treatment.

**Abstract:**

B-cell chronic lymphocytic leukemia (B-CLL) is the most common type of leukemia in the Western world. Mutation in different genes, such as *TP53* and *ATM*, and deletions at specific chromosomic regions, among which are 11q or 17p, have been described to be associated to worse disease prognosis. Recent research from our group has demonstrated that, contrary to what is the usual cancer development process through missense mutations, B-CLL is driven by the overexpression of the small GTPase *RRAS2* in its wild-type form without activating mutations. Some mouse models of this disease have been developed to date and are commonly used in B-CLL research, but they present different disadvantages such as the long waiting period until the leukemia fully develops, the need to do cell engraftment or, in some cases, the fact that the model does not recapitulate the alterations found in human patients. We have recently described Rosa26-*RRAS2*^fl/fl^xmb1-Cre as a new mouse model of B-CLL with a full penetrance of the disease. In this work, we have validated this mouse model as a novel tool for the development of new therapies for B-CLL, by testing two of the most broadly applied targeted agents: ibrutinib and venetoclax. This also opens the door to new targeted agents against R-RAS2 itself, an approach not yet explored in the clinic.

## 1. Introduction

B-cell chronic lymphocytic leukemia (B-CLL) is the most frequent type of leukemia in the Western World [1], with an incidence per year of 3.1 and 6.1 per 100,000 in females and males, respectively [2]. The 5-year survival rate has increased from 77.3% in 2009 to 87.9% as of today, thus highlighting the improvements in treatment options and diagnosis [3,4]. The median age of B-CLL diagnosis is 72 years of age [5,6]. This disease is characterized by the accumulation of CD5+ B lymphocytes in the blood and other lymphoid organs such as spleen and lymph nodes [7]. CD5 is a cell marker normally found in T cells that is characteristically expressed in leukemic B cells, but not in normal B cells [8]. Genes such as *TP53*, *ATM*, *MYD88*, others involved in Notch signaling, inflammatory pathways, B cell receptor signaling and others have been described to be involved in B-CLL development [9], as well as genes regulating the MAPK-ERK and MYC pathways [10]. Recurrent deletions in chromosomes 8p, 11q, 13q and 17p and trisomy of chromosome 12 have also been described [9]. A study published in 2022 identified 82 additional putative drivers mutated in low frequencies (<2% of the analyzed patients) [11]. It is important to highlight that the alteration frequency of the mutated genes described to date mainly affects less than 10% of B-CLL patients, thereby suggesting that other alterations are likely to contribute to disease development.

Several prognostic biomarkers are well established nowadays to assess disease prognosis. Some are host factors, like gender and age, while others are cell marker expression levels (CD38, ZAP70, and CD49d), serological values (β2-microglobulin, LDH) and genetic alterations (deletion of chromosome arms 11q, 13q, 17p, *TP53* gene mutation and trisomy 12) and the mutational status of the *IGHV* gene [12,13,14].

First-line therapy has normally been based on chemoimmunotherapy, FCR (fludarabine, cyclophosphamide and rituximab), chlorambucil, or bendamustine with anti-CD20 antibodies (rituximab, obinutuzumab). Nevertheless, good and general long-term disease control with FCR is achieved only in patients with mutated *IGHV* genes. Treatment options in recent years are shifting thanks to the development of different inhibitors targeting kinases associated to BCR signaling, such as ibrutinib (Bruton’s tyrosine kinase (BTK)) and idelalisib (phosphoinositide 3-kinase δ (PI3Kδ)) [15], second-generation more specific BTK inhibitors (acalabrutinib), and venetoclax (a specific inhibitor of the anti-apoptotic factor BCL2), alone or in combination with anti-CD20 antibodies [16]. These more targeted clinical options have improved patient outcome compared to traditional chemotherapy. One example of this is the treatment with venetoclax, that induces cell death independently of *TP53* mutations and/or del(17p) [17]. Ibrutinib has been widely successful in the treatment of B-CLL and is currently the go to treatment option alone or in combination with other chemotherapeutic or immunotherapeutic agents [18].

The most common and popular mouse model of B-CLL is the Eμ-TCL1, where the T-cell leukemia oncogene *TCL1* is inserted under the control of the immunoglobulin heavy chain variable region promoter and immunoglobulin heavy chain enhancer (Eμ). This leads to the development of a B-CLL-like disease at a late age (13 to 18 months) [19]. The negative aspects of this animal model lay on the long period required to reach a fully developed form of the disease, and on its inability to represent a recurrent human B-CLL disease. Another approach has been the engraftment of the B-CLL cell line MEC-1 in *Rag2*^−/−^*γc*^−/−^ mice, which mimicked aggressive human B-CLL but it had the caveat that the MEC-1 cell line does not express CD5, the canonical marker expressed in human B-CLL patients [20]. Engraftment and growth of B-CLL patient-derived B cells injected intravenously in NSG mice has been successful and optimized upon co-injection with polyclonally-activated autologous T cells pre-stimulated in vitro [21]. However, in these experiments, the maximum window in which B-CLL biology and potential treatment avenues can be tested was of only 63 days [21]. Deletion of the *DLEU2/miR-15a/16-1* locus in mice, encoded in the chromosome arm 13q, leads to an indolent disease that recapitulates the phenotype observed in human B-CLL, but with a penetrance of the disease in these mice of approximately 50% [22]. These and other approaches to the generation of mouse models of B-CLL are thoroughly reviewed in [23]. These models are all currently used in B-CLL research, but newer approaches that recapitulate the full heterogeneity of B-CLL biology are needed. Henceforth, it is crucial towards mouse model development to prioritize full B-CLL penetrance, eliminating the need of xenografts, as well as, especially, promoting the development of the disease in the early stages of the mouse life. The latter characteristic ensures it will not be necessary to wait several months or even more than a year until being able to study this disease.

The RAS protein family comprises many small guanosine triphosphate hydrolases (GTPases), some of which have been widely found to be mutated and to be responsible for different types of human cancer [24]. The three classical members, K-RAS, H-RAS, and N-RAS were discovered and identified in the late 1960s, 1970s, and early 1980s [25]. RAS related proteins (R-RAS) share the GTP-binding domain, the effector switch I and II regions, the Raf binding domain, and the CAAX box in their C-terminus with the classical RAS proteins. They also share the factors that mediate their activation and inactivation cycles, guanine exchange factors (GEF) and GTPase activating proteins (GAP) [26]. Mutation of R-RAS2 in analogous residues to G12V and Q61L in classical RAS proteins induced comparable cell transformation in culture, as well as the growth of progressive tumors in mice [27,28]. A key difference between R-RAS2 and the classical RAS proteins is the high intrinsic nucleotide exchange activity of R-RAS2. It is able to exchange GDP for GTP without the cooperation of any specific GEF proteins at rates similar to those observed with H-RAS after addition of a specific GEF [29].

R-RAS2 has been described to be involved in different functions in the organism. It regulates platelet activation by means of its interaction with the glycoprotein VI-ITAM-containing collagen receptor [30]. In the central nervous system, R-RAS2 regulates Schwann cell migration [31]. Additionally, R-RAS2 also controls proper mammary gland development [32]. R-RAS2 protein has been found overexpressed in diverse cancer types such as oral squamous cell carcinoma [33], esophageal tumors [34], hepatocellular carcinoma, [35] and highly aggressive skin cancer [36]. When it is mutated, R-RAS2 induces primary breast tumorigenesis as well as late-stage metastasis [37]. Both mutation and overexpression of *RRAS2* caused transformation of breast cancer cell lines [38]. R-RAS2 has been found bearing activating mutations (Q72L and others) in patients with Noonan Syndrome [39,40]. Recent reports have shown that R-RAS2 harboring the Q72L mutation is a potent oncogenic driver that triggers the formation of a wide variety of tumors (ovarian cystadenomas, T-ALL, etc.) [41]. However, it is not only Q72L but also other mutations (G23V/A/C/S, G24D/C/V, A70T and Q72H) found in human cancer that have transforming potential when they are expressed in immortal cell lines [42].

Previous research from our groups showed that R-RAS2 interacted directly with both the B and T cell receptors (BCR and TCR) through their immunoreceptor tyrosine-based activation motif (ITAM), preferentially in the inactive GDP-bound form of R-RAS2. There, it provides tonic survival signals [43]. R-RAS2 also regulates the internalization of the TCR after immune synapse formation, only in its wild-type (WT) form [44]. In B cells, R-RAS2 regulates the correct formation of germinal centers via control of the B cell metabolism [45]. We have recently described, through the analysis of both human samples from B-CLL patients and our Rosa26-*RRAS2*^fl/fl^-mb1-Cre and Sox2-Cre mouse models, that overexpression of *RRAS2* drives B-CLL development [46]. In this work, we have treated Rosa26-*RRAS2*^fl/fl^-mb1-Cre mice with the drugs currently used in the clinic ibrutinib and venetoclax [16] and observed that, especially in the case of ibrutinib, the leukemic cell population recedes. This opens the door to the use of this mouse model as a tool to test new therapeutic avenues for B-CLL.

## 2. Materials and Methods

### 2.1. Mice

The Rosa26-*RRAS2^fl/fl^* knock-in mouse line was established on a C57Bl/6J genetic background. Briefly, it was generated by cloning the coding sequence for human R-RAS2 tagged with HA into the CTV vector (a gift from Klaus Rajewsky; Addgene plasmid #15912; http://n2t.net/addgene:15912 (accessed on 27 April 2007); RRID:Addgene_15,912) [47]. This construct was then inserted into the Rosa26 locus via homologous recombination using a genOway defined protocol. The detailed composition of the inserted cassette is explained in [46]. The correct insertion of the *RRAS2* expressing cassette was checked following the PCR screening strategy explained in [48]. This mouse line was crossed with mb1-Cre mice. These mice express the Cre recombinase specifically in B cells starting at an early precursor phase, since the *mb1* gene encodes the Igα signaling subunit of the BCR. Thereby, we achieve B-cell specific overexpression of *RRAS2*. Previously described mb1-Cre transgenic mouse lines were generously provided by Prof. Dr. Michael Reth (University of Freiburg, Germany) [49]. All mice were maintained under SPF conditions at the animal facility of the Centro de Biología Molecular Severo Ochoa (CBMSO) in accordance with national and European guidelines. All the procedures were approved by the ethical committee of the CBMSO and were under the Community of Madrid authorization numbers PROEX 384/15 and PROEX 296.7/21.

### 2.2. Cell Preparation

Spleens from mice were homogenized with 40 μm strainers (Falcon, Corning Incorporated, Corning NY, USA) and washed in phosphate-buffered saline (PBS) containing 2% FBS. Bone marrows were extracted from the tibias of mice by removing the proximal tibia and centrifuging at maximum speed for 30s. Blood was extracted from the facial vein via puncture and the blood was kept anticoagulated in an excess of 30 μL of heparin (1000 UI/mL) (Chiesi España, Barcelona, Spain). Spleen, bone marrow and blood cells were resuspended for 5 min in ACK buffer (0.15 M NH4Cl, 10 mM KHCO3, 0.1 mM EDTA, pH 7.2–7.4) to lyse and discard the erythrocytes and washed in PBS with 2% FBS. Cells were subsequently analyzed via flow cytometry after staining with the appropriate fluorescently labelled antibodies.

### 2.3. Mouse Drug Treatment

Twenty-three- to twenty-seven-week-old Rosa26-*RRAS2*^fl/fl^ xmb1-Cre mice were treated with either vehicle, ibrutinib at 25 mg/Kg or venetoclax at 50 mg/Kg for 31 days. Drugs were dissolved in a solution of 4% DMSO (Sigma–Aldrich, Merck KGaA, Darmstadt, Germany), 80% Kollisolv^®^ PEG E 400 (Sigma–Aldrich), 4% Tween 20 (Sigma–Aldrich) and 12% saline solution. 200 μL were administered via oral gavage every day to each mouse of 25 g of weight, adjusting the volume for heavier or lighter mice. Mice were bled at the start, the end, and halfway through the experiment from the facial vein and cells were stained and analyzed via flow cytometry to track the evolution of the tumoral cells. At the experiment endpoint, all mice were euthanized via CO_2_ inhalation and spleen and bone marrow were extracted. Bone marrow and blood were used in their entirety for flow cytometry analysis and the spleens were cut in half so that one part could be used for cytometry analysis and the other half, for hematoxylin/eosin staining.

### 2.4. Flow Cytometry

Mouse and human single-cell suspensions were incubated with Ghost Dye 540 (TONBO Biosciences, Thermo Fisher Scientific, Waltham MA, USA) at 1:500 dilution for 15 min in PBS to label and discard dead cells. After that, cells were incubated with fluorescently labelled antibodies for 30 min at 4 °C after blocking FC receptors using anti-CD16/32 antibody (1:250) for 15 min at 4 °C. Both of these steps are carried out in PBS + 2% FBS. The utilized fluorophore-labelled antibodies were anti-CD5-PE, anti-CD19-PECy7, anti-IgM-APC, anti-B220-APCCy7 and anti-IgD-eF450, all from BD Pharmingen (BD Biosciences, Franklin Lakes, NJ, USA). Afterwards, cells were washed in PBS + 2% FBS and data were collected on a FACS Canto II (BD Biosciences) cytometer. A minimum of 50,000 and a maximum of 200,000 events was acquired in every measurement. Analyses were performed using FlowJo v10 software (TreeStar, Ashland, OR, USA, BD Biosciences). Counting of total cells was performed with CountBright™ beads (Invitrogen, Waltham, MA, USA, Thermo Fisher Scientific).

### 2.5. Hematoxylin and Eosin Staining

Spleens from Rosa26-*RRAS2*^*fl*/*fl*^xmb1-Cre mice were fixed in formalin solution, neutral buffered, 10% (Sigma) overnight at 4 °C immediately after euthanizing the mice. They were then washed with PBS and sent to the Histology service at the Centro Nacional de Biotecnología (Madrid, Spain), where the hematoxylin/eosin staining was carried out. Images of the stainings were captured using a vertical AxioImager M1 microscope (Carl Zeiss AG, Oberkochen, Germany). Follicle areas were calculated using the formula π × a × b, being a and b the two radii of the ellipse. All follicles visible in their entirety were used to calculate the areas in three representative images per mouse.

### 2.6. Antibodies and Western Blotting

To analyze whole-cell lysates in Western blots, the cells were lysed in Brij96 lysis buffer with protease and phosphatase inhibitors (0.5% Brij96, 140 mM NaCl, 20 mM Tris-HCl (pH 7.8), 10 mM iodoacetamide, 1 mM phenylmethylsulfonyl fluoride (PMSF), leupeptin (1 µg/mL), aprotinin (1 µg/mL), 1 mM sodium orthovanadate and 20 mM sodium fluoride), resolving the lysates by SDS-PAGE and transferring the proteins to nitrocellulose membranes using a semi-dry transfer procedure (Trans Blot Turbo, Biorad Laboratories Inc., Hercules, CA, USA). The membranes were blocked for 1 h in 5% BSA (Sigma-Aldrich) in TBS-T (25 mM Tris-HCl [pH 8.0], 150 mM NaCl, 0.1% Tween-20) and then incubated overnight at 4 °C with the appropriate primary antibodies diluted in blocking buffer. After three washes with TBS-T, the membrane was incubated for 45 min at rt with the secondary antibody (1:30,000 dilution: Jackson Immunoresearch Laboratories, West Grove, PA, USA) and antibody binding was detected via standard chemoluminescence with a Kodak X-OMAT 2000 Processor (Eastman Kodak Company, Rochester, NY, USA). Antibodies used were, HA influenza hemagglutinin epitope (#12CA5, Sigma-Aldrich); R-RAS2 (#H00022800-M01, Abnova Gmbh, Taipei, Taiwan); panRAS (#05-516, Merck Millipore, Burlington, MA, USA); vinculin (#ab129002, Abcam Corporate, Cambridge, UK).

### 2.7. Statistical Analysis

Statistical parameters including the exact value of n, the mean ± S.E.M. are described in the Figure 1, Figure 2, Figure 3 and Figure 4 and Figure legends. Two-tailed unpaired *t*-test with Welch’s correction and one-way ANOVA tests were used as indicated to assess the significance of mean differences. The number of mice to be used for comparison was calculated from preliminary experiments aimed to generate significant data using a two-sided *t*-test with alpha = 0.05 and a standard deviation of about 0.3. Outliers for the different analyses were identified using ROUT model at Q = 1% to remove definite outliers from analysis. All data were analyzed using the GraphPad Prism 10 software (GraphPad Company, Boston, MA, USA).

## 3. Results

### 3.1. Rosa26-RRAS2fl/flxmb1-Cre Mice Express Twice as Much R-RAS2 Protein in the Spleen as Their Wild-Type Counterparts

We have previously described a mouse model of CLL resulting from the overexpression of the human RRAS2 gene inserted into the Rosa26 locus of C57BL/6 mice [46]. The human RRAS2 gene was flanked by LoxP sites so that recombination with Cre recombinase would lead to the loss of an upstream STOP codon and expression of the protein. Rosa26-*RRAS2*^fl/fl^ mice were crossed with mb1-Cre mice to induce recombination and overexpression of RRAS2 specifically in B cells. The human R-RAS2 protein was engineered to express an HA epitope tag at the N-terminal end, allowing for distinction from the endogenous protein.

To measure the level of protein overexpression, we quantified it through Western blot analysis of total spleen cell lysates. Using an anti-HA epitope antibody, we specifically detected the expression of the human R-RAS2 protein in the spleens of Rosa26-*RRAS2*^fl/fl^xmb1-Cre mice but not in Rosa26-*RRAS2*^fl/fl^ or non-transgenic C57BL/6 mice (Figure 1a). After immunoblotting with an anti-R-RAS2 monoclonal antibody, we observed the presence of the HA-R-RRAS2 protein slightly above the size of the endogenous mouse R-RAS2 protein, which was present in all spleen samples. This second Western blot allowed us to calculate the expression of the human HA-tagged protein as approximately equal to that of the endogenous protein. Incubation with a pan-RAS antibody and with anti-vinculin served as controls for loading (Figure 1a). These results indicate that Rosa26-*RRAS2*^fl/fl^xmb1-Cre mice express approximately twice the amount of R-RAS2 protein compared to that found in control mice.

**Figure 1 cancers-15-05817-f001:**
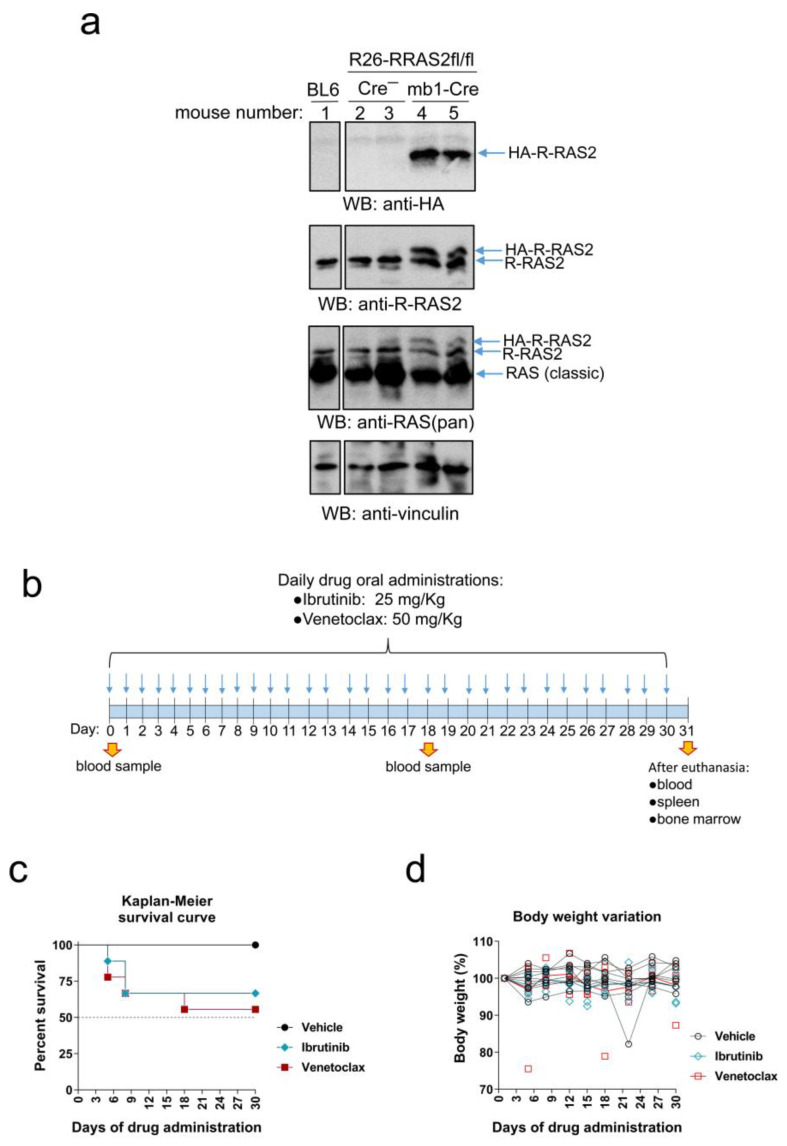
Rosa26-*RRAS2*^fl/fl^xmb1-Cre mice can go through a one-month long treatment with ibrutinib or venetoclax. (**a**) Western blot analysis of R-RAS2 protein expression in detergent lysates of total spleen cells from mice of the indicated genotypes. The positions of molecular weight markers are indicated to the left of each blot. (**b**) Schematic representation of the treatment protocol used in Rosa26-*RRAS2*^fl/fl^xmb1-Cre mice. (**c**) Kaplan–Meier survival curve of the mice under treatment. (**d**) Relative percentage of mouse weight evolution through the course of the treatment. *n* = 9 mice per group started the experiment. Full pictures of the Western blots and the densitometry scans are presented in Appendix A.

### 3.2. Testing the Rosa26-RRAS2^fl/fl^xmb1-Cre Mouse Model of CLL in Response to Ibrutinib and Venetoclax

To test the validity of the Rosa26-*RRAS2*^fl/fl^xmb1-Cre mouse model of spontaneous development of B-CLL for preclinical testing of new compounds, we subjected those mice to treatment with two of the drugs most commonly used in molecular targeted therapies for B-CLL: ibrutinib [15,16] and venetoclax [16,17]. To this end, we created three groups of 23- to 27-week-old Rosa26-*RRAS2*^fl/fl^xmb1-Cre mice, with nine mice each with comparable means of circulating leukemic CD19+CD5+ cells. These mice were administered daily single doses of ibrutinib, venetoclax or just vehicle by oral gavage as indicated in the cartoon of Figure 1b. Blood samples were taken at an intermediate time point (18 days of treatment) to analyze the effect of the drugs on the circulating leukemic cells. All animals were euthanized at day 31 to investigate the effect of drug treatment on leukemic cells in blood and also spleen and bone marrow. Both drug administration schedules resulted toxic to some animals since they caused the mortality of three of nine mice (Ibrunitib) and four of nine mice (Venetoclax) by day 18. No drug-related deaths were registered afterwards, up to day 31 (Figure 1c). In the rest of the treated mice, there was no harmful effect in respect to their body weight (Figure 1d). Nevertheless, general evaluation of the general aspect of these mice revealed that those under ibrutinib treatment, and more evidently under venetoclax, became mildly lethargic upon treatment progression as compared to those that were given vehicle alone.

### 3.3. Both Ibrutinib and Venetoclax Reduced Splenomegaly in Rosa26-RRAS2^fl/fl^xmb1-Cre Mice

The effect of drug treatment for 31 days was evaluated in lymphoid organs after euthanasia. We found that both drug treatments reduced spleen weight in a significant manner (Figure 2a). This is relevant, since we have previously described that *RRAS2* overexpression in B cells led to splenomegaly [46]. We observed that venetoclax treatment led to a significant decrease in the number of total CD19+ B cells. However, ibrutinib treatment did not affect the number of total B cells in the spleen (Figure 2b). Next, we examined the effect of drug treatment on leukemic B cells according to the expression of the characteristic marker CD5; we found no significant reduction by any of the two treatments on the percentage and number of CD19+CD5+ leukemic B cells (Figure 2c), although there was a partial reduction of this population with ibrutinib. The construct used to insert the human *RRAS2* gene in the Rosa26 locus of mice has an IRES sequence that allows the co-transcription of the green marker GFP together with R-RAS2 [46]. Therefore, GFP expression is a marker of *RRAS2* overexpression in B cells. Only the venetoclax treatment reduced the percentage and number of GFP+ CD19+ B cells, even though this difference was only significant for cell numbers. Meanwhile, ibrutinib treatment had no effect in the abundance of GFP+ CD19+ cells (Figure 2d). Leukemic B-CLL B cells express both membrane IgM and IgD as their B cell antigen receptors [50]. Nevertheless, the expression of IgM is higher than that of IgD [51], especially in patients with unmutated *IGHV* genes, those with a worse disease prognosis [50]. We found that venetoclax, but not ibrutinib, treatment provoked a significant reduction in the number of follicular B cells in Rosa26-*RRAS2*^fl/fl^xmb1-Cre mice (Figure 2e), suggesting a toxic effect of venetoclax on a healthy B cell population. Provided that ibrutinib and venetoclax reduced the splenomegaly of Rosa26-*RRAS2*^fl/fl^xmb1-Cre mice (Figure 2a), we studied if both treatments had an effect on the size of spleen follicles since these are the places for normal and malignant B cell location and maturation. As previously described by our group [46], we found in spleen sections stained with hematoxylin and eosin that control Rosa26-*RRAS2*^fl/fl^xmb1-Cre mice treated with just the vehicle had abnormally large follicles (Figure 2f). A quantitation of the area occupied by each follicle in different sections of the spleens showed that both ibrutinib and venetoclax treatments resulted in a significant reduction of the follicle size, being the effect of venetoclax more potent (Figure 2g). Altogether, these results show that both ibrutinib and venetoclax reduced splenomegaly and the size of spleen follicles but do not have a significant effect on the frequency and size of the leukemic CD19+CD5+ population in the spleen. In addition, venetoclax seems to have a toxic effect on non-leukemic follicular B cells.

**Figure 2 cancers-15-05817-f002:**
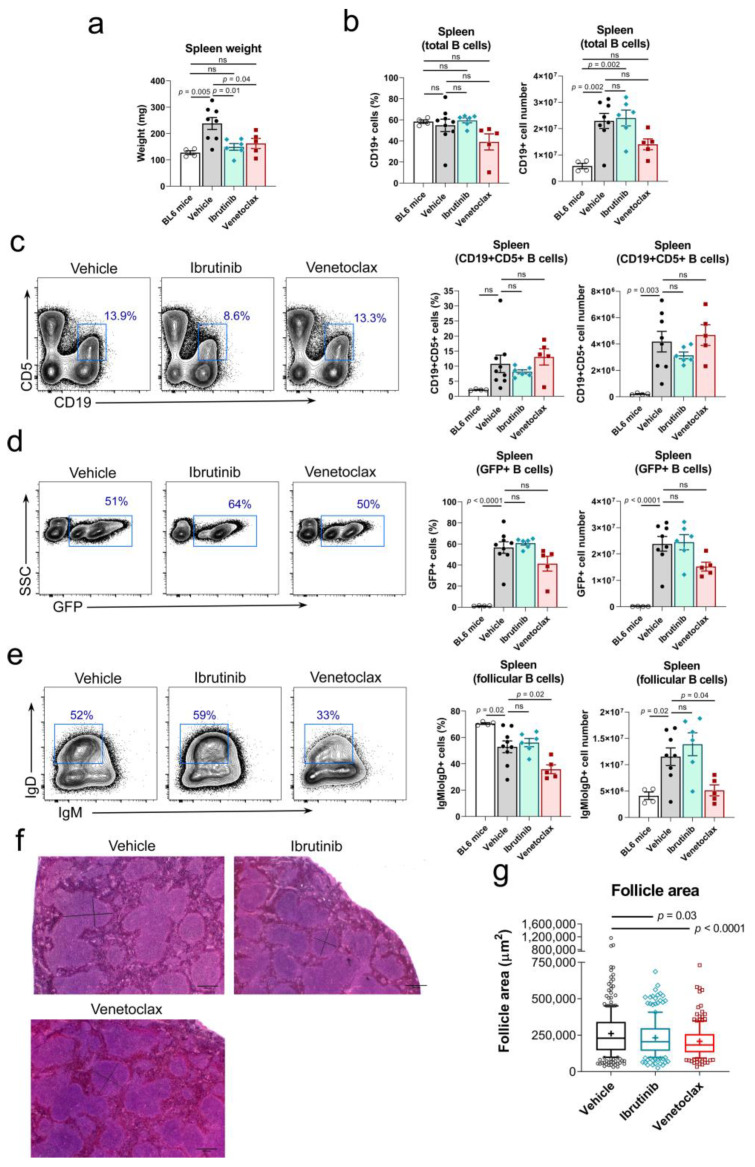
Ibrutinib and venetoclax treatment reverse the splenomegaly and follicle enlargement induced by *RRAS2* overexpression. (**a**) Spleen weights of mice in the vehicle, ibrutinib and venetoclax treatment groups at the experiment endpoint, day 31. Two-tailed unpaired *t*-test with Welch’s correction. (**b**) Quantification of the number of CD19+ B cells in the spleens of the treated mice at the experiment endpoint. Two-tailed unpaired *t*-test with Welch’s correction. Left, percentage of the total lymphocyte population; right, total numbers. (**c**) Left, representative two-parameter flow cytometry plots of CD19 and CD5 expression in the spleens of mice in each of the three established groups. The CD19+CD5+ population is highlighted with a blue box. Right, quantification of the number of CD19+CD5+ B cells in the spleens of the treated mice at the experiment endpoint. Left graph, percentage of the total lymphocyte population; right graph, total numbers. Two-tailed unpaired *t*-test with Welch’s correction. (**d**) Left, representative two-parameter flow cytometry plots of GFP expression vs. side scatter (SSC) in the spleens of mice in each of the three established groups. The GFP+ population is highlighted with a blue box. Right, quantification of the number of GFP+ cells in the spleens of the treated mice at the experiment endpoint. Left graph, percentage of the total lymphocyte population; right graph, total numbers. Two-tailed unpaired *t*-test with Welch’s correction. (**e**) Left, representative two-parameter flow cytometry plots of IgM and IgD expression within the total CD19+ B cell population. The follicular IgMlowIgD+ population is highlighted with a blue box. Right, quantification of the number of IgMlowIgD+ cells with the CD19+ gate in the spleens of the treated mice at the experiment endpoint. Left graph, percentage of the total CD19+ population; right graph, total numbers. Two-tailed unpaired *t*-test with Welch’s correction. (**f**) Representative hematoxylin and eosin stainings of the spleens of mice in the vehicle, ibrutinib and venetoclax groups at the experiment endpoint. Scale bars represent 400 μm. In each of the three images, the black bars are illustrative of the diameters used to calculate follicle areas. (**g**) Box and whisker plot showing all points and median value of the quantification of follicle areas using the diameters illustrated in the images in (**f**). Areas were calculated for all follicles visible in their entirety in three representative images per mouse. In all panels, *n* = 9 in the vehicle group, *n* = 6 in the ibrutinib group, *n* = 5 in the venetoclax group. Two-tailed unpaired t-test with Welch’s correction. ns: not significant.

### 3.4. Venetoclax but Not Ibrutinib, Reduced the Number of Non-Leukemic B220^high^IgM+ Immature B-Cells Precursors in the Bone Marrow

We also analyzed the bone marrow for the presence of leukemic cells, seeking for an effect of the two drug treatments. We found that ibrutinib did not significantly reduce the number of GFP+ CD19+ B cells (Figure 3a) or the number of CD19+CD5+ B cells in the bone marrow (Figure 3b), suggesting that this drug did not have an impact on the infiltration of the bone marrow by leukemic cells. In the case of venetoclax, there was a small reduction in the number of both GFP+ CD19+ B cells (Figure 3a) and CD19+ CD5+ B cells (Figure 3b), although this difference is not significant. The effect of both drug treatments on the generation of B cell precursors in the bone marrow according to the expression of the B220 and IgM markers showed that venetoclax treatment produced a significant reduction in the number of immature B220^high^IgM+ B cells but not in the number of B220^int^IgM^—^ pro-pre-B-cells (Figure 3c), suggesting the existence of a partial blockade on normal pro-pre-B-cell to immature B cell differentiation in the bone marrow. Opposed to this, ibrutinib treatment did not have any deleterious effect in the B-cell maturation process in the bone marrow (Figure 3c). The B-cell maturation blockade induced by venetoclax may be behind the partial reduction of GFP+ CD19+ and CD19+ CD5+ cells observed in the bone marrow (Figure 3a and Figure 3b, respectively), as well as the decrease of total CD19+ and GFP+ B cells in the spleen (Figure 2b and Figure 2d, respectively).

**Figure 3 cancers-15-05817-f003:**
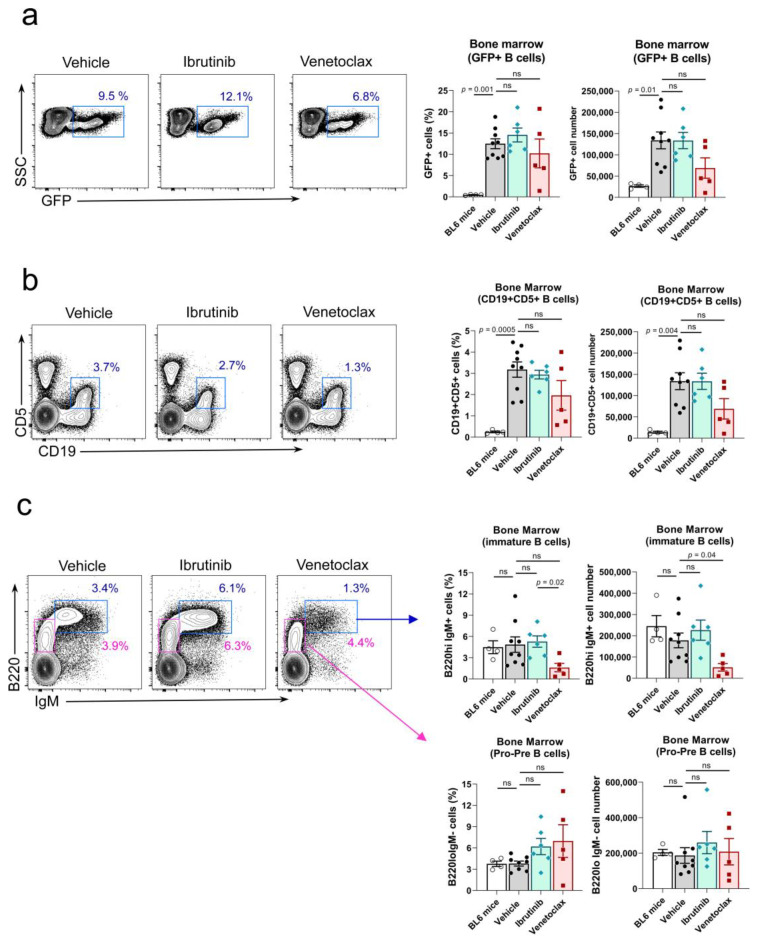
Venetoclax treatment hampers transition from pro-pre-B to immature cell stage in the bone marrow. (**a**) Left, representative two-parameter flow cytometry plots of GFP expression vs. side scatter (SSC) in the bone marrow of mice in each of the three established groups. The GFP+ population is highlighted with a blue box. Right, quantification of the number of GFP+ cells in the bone marrow of the treated mice at the experiment endpoint. Left graph, percentage of the total lymphocyte population; right graph, total numbers. Two-tailed unpaired *t*-test with Welch’s correction. (**b**) Left, representative two-parameter flow cytometry plots of CD19 and CD5 expression in the bone marrow of mice in each of the three established groups. The CD19+CD5+ population is highlighted with a blue box. Right, quantification of the number of CD19+CD5+ cells in the bone marrow of the treated mice at the experiment endpoint. Left graph, percentage of the total lymphocyte population; right graph, total numbers. Two-tailed unpaired *t*-test with Welch’s correction. (**c**) Left, representative two-parameter flow cytometry plots of IgM and B220 expression within the total CD5− population in the bone marrow of mice in each of the three established groups. The immature B220hi IgM+ population is highlighted with a blue box. The pro-pre-B-cell population is highlighted with a pink box. Right, top graphs, quantification of the number of immature B cells in the bone marrow of the treated mice at the experiment endpoint. Left graph, percentage of the total CD5− population; right graph, total numbers. Right, bottom graphs, quantification of the number of pro-pre-B-cells in the bone marrow of the treated mice at the experiment endpoint. Left graph, percentage of the total CD5− population; right graph, total numbers. In all panels, *n* = 9 in the vehicle group, *n* = 6 in the ibrutinib group, *n* = 5 in the venetoclax group. Two-tailed unpaired *t*-test with Welch’s correction. ns: not significant.

### 3.5. Ibrutinib, but Not Venetoclax, Treatment Reduced the Percentage of Malignant CD5+ B Cells in Blood of Rosa26-RRAS2^fl/fl^xmb1-Cre Mice

The progressive efficacy of ibrutinib and venetoclax treatments was also analyzed in peripheral blood while the treatment regime was ongoing at day 18 and at the experiment endpoint at day 31. Even though venetoclax treatment had a tendency to reduce the total number of B cells at both time points, this difference was not significant (Figure 4a). This phenotype is probably a reflection of the pro-pre-B to immature B cell maturation blockade observed in the bone marrow after venetoclax treatment (Figure 3c). The analysis of the number of leukemic B cells according to the expression of the CD5 marker by CD19+ B cells (Figure 4b) did not show significant effects when data were analyzed in bulk (Figure 4c). However, if the analysis is focused within the total CD19+ B cell gate (Figure 4d), we observed significant differences in the behavior of leukemic B220^low^CD5+ and normal, healthy B220+CD5− B cells. In the case of the leukemic B220^low^CD5+ cells, we observed that the variation of this population in the ibrutinib-treated mice was opposite to that of venetoclax-treated or the control group. Already at day 18, the percentage of leukemic cells of mice treated with ibrutinib decreases compared to the beginning of the experiment, whereas both in the venetoclax-treated mice and those only given vehicle show an increase in this population. This effect is further enhanced by day 31 (Figure 4e). In this analysis, the effect achieved with ibrutinib treatment is only significantly different to the mice treated with venetoclax due to the partial heterogeneity at the starting point of the procedure (Figure 4e). If in all three cases the starting point is set as a 100% and we measure the relative evolution of the B220^low^CD5+ cells, we can observe more clearly the different dynamics of mice treated with ibrutinib as compared to venetoclax and vehicle (Figure 4f). Confirming the specific effect of ibrutinib on circulating B cells, we observed an opposite effect in the normal B220+CD5− normal B cell population. Ibrutinib led to a progressive increase in the percentage of this population, whereas mice administered venetoclax or vehicle solution showed a reduction in these cells (Figure 4g). Coupled with the results obtained in the spleen (Figure 2e), we observed that ibrutinib treatment provoked a significant increase in the normal follicular IgMlow IgD+ population in the blood, while venetoclax had no impact in this characteristic (Figure 4h).

**Figure 4 cancers-15-05817-f004:**
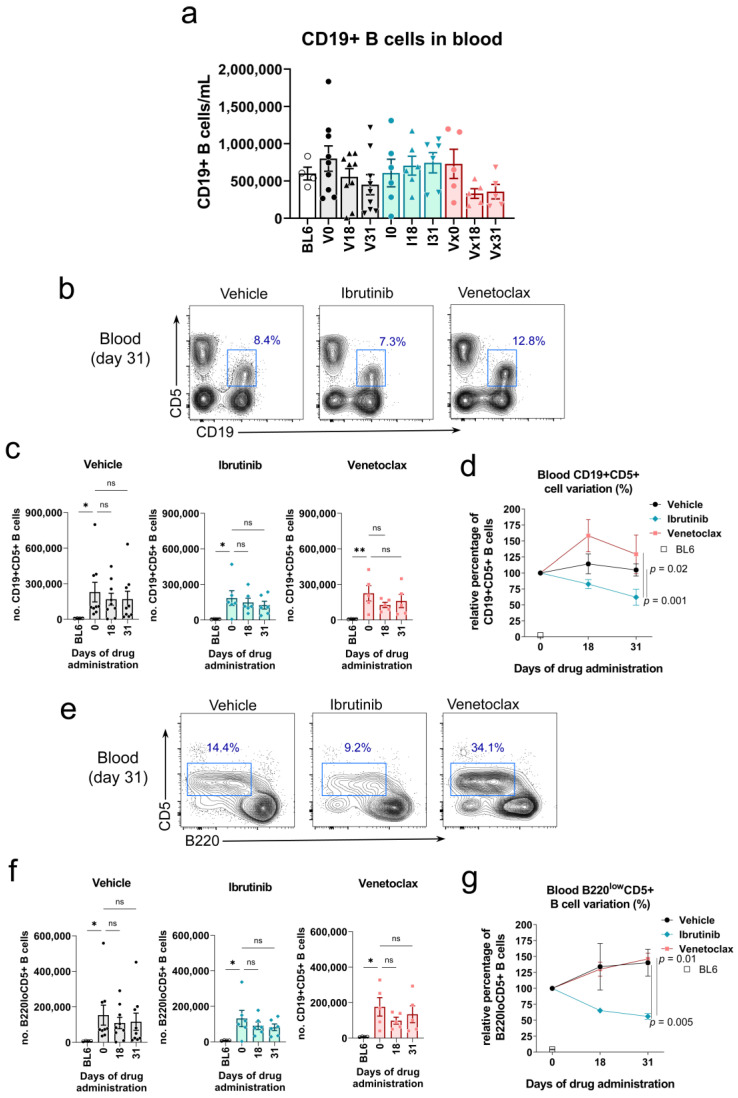
Ibrutinib treatment reduces the circulating CD19+B220^low^CD5+ leukemic population in the blood of Rosa26-*RRAS2*^fl/fl^xmb1-Cre mice. (**a**) Quantification of the total number of CD19+ B cells in the blood of mice at the start (0), halfway through the experiment (18) and the experiment endpoint (31) in the vehicle, ibrutinib and venetoclax groups. Column bars represent mean values ± SEM. (**b**) Representative two-parameter flow cytometry plots of CD19 and CD5 expression in the blood of mice in each of the three established groups at the experiment endpoint. The CD19+CD5+ population is highlighted with a blue box. (**c**) Quantification of the total number of CD19+CD5+ leukemic B cells in the blood of mice at the start (0), halfway through the experiment (18) and at the experiment endpoint (31) in the vehicle, ibrutinib and venetoclax groups. *, *p* = 0.02; **, *p* = 0.01 (Kruskall-Wallis test. (**d**) Representative two-parameter flow cytometry plots of B220 and CD5 expression within the total CD19+ population at the endpoint of the experiment in the blood of the three established groups. The B220^low^CD5+ population is highlighted with a blue box. (**e**) B220^low^CD5+ leukemic cells evolution, within the total CD19+ gate, in the blood of treated mice. Two-way ANOVA test. (**f**) B220^low^CD5+ population shown in (**e**) relative evolution through the course of treatment. Two-way ANOVA test. (**g**) B220+CD5− cells evolution, within the total CD19+ gate, in the blood of treated mice. *, *p* = 0.02 (Kruskall-Wallis test).

## 4. Discussion

There are not many mouse models of B-CLL that fully recapitulate the disease. The most commonly used one is the Eμ-TCL1. In this model, the leukemic CD19+CD5+ cells are detected in the spleen after 3 to 5 months and the blood disease only appears after more than one year of age (13 to 18 months) [19]. In addition to that, it requires engineering the T-cell leukemia oncogene TCL1 under the control of the immunoglobulin heavy chain variable region promoter and immunoglobulin heavy chain enhancer (Eμ). Other options to establish a mouse model of B-CLL include carrying out a xenograft with either the B-CLL cell line MEC-1 [20] or with patient-derived B-CLL B cells together with autologous, in vivo pre-stimulated T cells [21]. Additionally, deletion of the *DLEU2/miR-15a/16-1* locus in mice recapitulates an indolent B-CLL disease [22]. The Rosa26-*RRAS2*^fl/fl^xmb1-Cre mouse line reported in [46] is a novel model of B-CLL where the leukemic CD19+CD5+ cells appear early in the life of mice. It presents a 100% penetrance of the disease and manifests concurrently with splenomegaly and a progressive blood lymphocytosis that leads to a reduced lifespan of the mice [46]. In this work we have validated the use of the Rosa26-*RRAS2*^fl/fl^xmb1-Cre mouse model as a tool for the preclinical testing of new therapeutic options for B-CLL. Treatment of the Rosa26-*RRAS2*^fl/fl^xmb1-Cre mice with the drugs currently used in the clinic venetoclax and, especially, ibrutinib leads to a reduction of the CD19+B220loCD5+ leukemic population in the blood. A similar trend with this population is observed in the spleen. The fact that three mice from the ibrutinib arm and four from the venetoclax treatment had died by day 18 probably reflects the side effects that these therapies have on the organism, also in some human B-CLL patients, because the rest of the treated animals remained in good health and general aspect. In line with this thought, one of the venetoclax-treated mice had to be euthanized due to a sudden weight loss and general worsening of its condition. Analysis of the blood, bone marrow and spleen of this mouse brought to light the fact that it had lost the vast majority of its lymphocytes. In accordance with this, we observed that venetoclax treatment leads to a severe reduction of the immature B-cell population in the bone marrow, which may explain the eventual loss of the majority of the B cell population in some cases. In turn, this described effect in the bone marrow is accompanied by a reduced CD19+ population in the spleen. This reflects that the venetoclax treatment is not selective for the leukemic cells. Severe side effects are also described in humans, where venetoclax can cause tumor lysis syndrome and secondary infections [52], as well as neutropenia, anemia or thrombocytopenia [53]. All this clearly calls for the development of new therapeutic regimens that maximize efficacy while minimizing their side effects. The splenomegaly observed in our Rosa26-*RRAS2*^fl/fl^xmb1-Cre mice mimics the alteration found in humans [5]. This disease characteristic is reverted after one month of ibrutinib and venetoclax treatment, which significantly reduced spleen weights. This treatment outcome reflects the one observed in B-CLL patients after ibrutinib [54,55] and venetoclax [56] therapy, as well as the effect of both ibrutinib and venetoclax treatment in the B-CLL model Eμ-TCL1 [57]. All this enhances the recapitulation of the human B-CLL disease and its response to treatment in our Rosa26-*RRAS2*^fl/fl^xmb1-Cre mouse model.

A characteristic that should be considered in future approaches for the optimization of this process is the evaluation of gender-related effects on treatment efficacy. In this work, the distribution of males and females was at a 2:1 ratio and mice were separated into the three treatment groups without taking this factor into consideration, solely based on the purpose of having comparable means of the leukemic CD19+CD5+ population at the start of treatment. It would be recommendable to study if the sex influence is an issue in treatment response in the Rosa26-*RRAS2*^fl/fl^xmb1-Cre model, since B-CLL is a disease that affects male patients approximately twice as much as females [2].

One treatment option that, as of yet, has not been explored for B-CLL or, incidentally, any other malignancy, is the use of direct inhibitors of R-RAS2. As demonstrated in [46], overexpression of WT *RRAS2* drives the development of B-CLL. Based on this, direct targeting of this protein is an attractive prospect that warrants further research. Direct inhibitors administered alone or in combination with other treatment options such as ibrutinib may increase the efficacy of the therapeutic regime and also decrease the probability of development of ibrutinib-resistance [58,59].

## 5. Conclusions

In sum, we have validated the use of Rosa26-*RRAS2*^fl/fl^xmb1-Cre mice as a novel model for preclinical testing of B-CLL drugs by means of the treatment with currently approved agents ibrutinib and venetoclax. This paves the way for the development of new drugs potentially targeting, among others, R-RAS2 itself, alone or in combination with other therapeutic agents.

## Data Availability

The data presented in this study are available upon request to the corresponding author.

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
