# Peer review of "Mice Overexpressing Wild-Type RRAS2 Are a Novel Model for Preclinical Testing of Anti-Chronic Lymphocytic Leukemia Therapies"

_cancers, 2023, doi:10.3390/cancers15245817_

Round 1

Reviewer 1 Report

Comments and Suggestions for Authors

The authors tested the Rosa26-RRAS2fl/fl knock-in mouse line as a model for B-cell chronic lymphocytic leukemia (B-CLL), thus potentially benefiting future B-CLL studies. Yet, some additions and further discussions are highly recommended for the ease of our future readers’ understanding. See the details below.

According to Subsection 2.3, 23- to 27-week-old mice were used. Does this address the need for a B-CLL mouse model having a shorter development period as introduced in Section 1? Also, Subsection 2.3 indicated blood drawing was performed at the beginning of the experiment. Accordingly, it is suggested to amend Figure 1a to reflect such a step.

As shown in Figure 1b and the related description, a significant percentage of mice in the treatment groups died. Does this result indicate the drug dosing is too high? Or, in other words, does this result indicate the current mouse model requires a very high dose to show the drug’s effect? This figure is also confusing in a way that, normally, the death of tested mice is expected in the diseased group, but not the treatment ones. Using this mouse model to test a potential drug in the future, do the authors suggest not using survival as a readout?

Concerning Figure 2 and the related description, it is suggested to include a further control not having splenomegaly, such as healthy mice at a similar age without any treatment or only treated with the vehicle. By adding this control, we will be able to not only confirm the splenomegaly successfully induced in this B-CLL model but also quantify how successful the treatments are. Similar comments apply to the data shown in Figures 3-4. Without a reference showing the healthy status, it is very hard to evaluate how a tested drug performs from the perspectives of efficacy and safety.

Further concerning Figures 2-3 and their related description, if a drug treatment (such as ibrutinib) does not show any decrease in the percentage/number of CD5+ B cells in the spleen or bone marrow (particularly considering the Introduction section noted that “B-CLL exhibits CD5 expression in leukemic B cells, a marker typically found in T cells but uniquely present in B-CLL cells, distinguishing them from normal B cells), does it raise a question of the effectiveness of such a drug?

Kindly confirm all panels of Figure 4 are showing up correctly.

Comments on the Quality of English Language

Some minor language changes are suggested.

(1)          It seems there are two Subsections 2.5.  In Section 3, only one subsection is included, which is 3.1. It is suggested to either remove the title of Subsection 3.1, or divide Section 3 into multiple subsections.

(2)          The sentence spanning lines 34-35 seems incomplete. Accordingly, it is suggested to change the “.” in line 35 to “,” instead. A similar comment applies to the first “.” in line 96.

(3)          The sentence fragment spanning lines 55-56 can be confusing. It is suggested to add “inhibiting” before “Bruton’s tyrosine kinase (BTK)” and “phosphoinositide 3-kinase (PI3K)δ.”

(4)          For consistency, shall we also italicize the gene name “TCL1” in lines 64 and 360? Kindly also verify the fonts of “-/-“ in line 69, “NH4Cl” and “KHCO3” in line 144, “IgMlowIgD+” in lines 249 and 250, “B220hi” in line 304, and “B220low” in the description of Figure 4.

(5)          The “are” in line 166 is suggested to be changed to “were.” 

Reviewer 2 Report

Comments and Suggestions for Authors

In this study, the authors utilized the Rosa26-RRAS2fl/flxmb1-Cre model to validate the effects of two drugs on B-CLL therapies. Treatment of these mice with ibrutinib and venetoclax resulted in a reduction in CD19+B220lowCD5+ leukemic cells in the blood and spleen. However, I have several questions that I would like the authors to address:

1: The description of mouse model construction is not sufficiently clear. The authors mentioned that they used "genOway technologies." This appears to be the name of a company rather than a description of the mouse construction technique. Please provide detailed information on how the mice were constructed and the breeding process.

2: Regarding the mouse construction process, the authors only referred to citation 46, and the supplementary material for citation 46 contains errors in the vector diagrams for the mice. Please provide accurate vector diagrams for the mouse construction in this section and provide evidence of successful mouse construction.

3: Please provide Western blot evidence to demonstrate the changes in RRAS expression before and after CRE expression. Include this data in Figure 1.

4: In Figure 1, please clarify whether the mice used in the study were single-copy or double-copy mice.

Reviewer 3 Report

Comments and Suggestions for Authors

The authors have earlier showed RRAS2 has a role in CLL development. In this study authors have tried treating Rosa26-RRAS2fl/fl-mb1 Cre  mice with the first line drugs Ibrutinib and Venetoclax. This is nice work however, the following comments will make the manuscript comprehensive.

1. The English and scientific language needs improvement.

2. Figures needs better resolution and making figures homogenous with better resolution will be better.

3. Have authors tried combination therapy (Ibrutinib and Venetoclax, together) and any data on that.

4. Figure 1 and 2 needs better labelling, N= number, p values and time.

Comments on the Quality of English Language

The English and scientific language needs improvement.

Round 2

Reviewer 1 Report

Comments and Suggestions for Authors

Thanks for the authors’ amendments and elaboration. I only have some minor suggestions.

In addition to the language changes and the section numberings mentioned in the previous round of discussion, the following are further noticed.

(1) There is an extra “)” in line 58.

(2) Kindly correct the title of the plots in Figure 2e from “Spleen weight” to “Spleen” if appropriate. 

Author Response

We thank the reviewer for the corrections to the manuscipt. We have now amended Figure 2 and corrected lane 58.
